# Effect of Sprint Interval Training on Cardiometabolic Biomarkers and Adipokine Levels in Adolescent Boys with Obesity

**DOI:** 10.3390/ijerph191912672

**Published:** 2022-10-03

**Authors:** Marit Salus, Vallo Tillmann, Liina Remmel, Eve Unt, Evelin Mäestu, Ülle Parm, Agnes Mägi, Maie Tali, Jaak Jürimäe

**Affiliations:** 1Institute of Sports Sciences and Physiotherapy, Faculty of Medicine, University of Tartu, Ujula 4, 51008 Tartu, Estonia; 2Department of Physiotherapy and Environmental Health, Tartu Health Care College, Nooruse 5, 50411 Tartu, Estonia; 3Department of Pediatrics, Institute of Clinical Medicine, Faculty of Medicine, University of Tartu, Lunini 6, 50406 Tartu, Estonia; 4Children’s Clinic of Tartu University Hospital, Lunini 6, 50406 Tartu, Estonia; 5Department of Sports Medicine and Rehabilitation, Institute of Clinical Medicine, Faculty of Medicine, University of Tartu, Puusepa 8, 50406 Tartu, Estonia; 6Sports Medicine and Rehabilitation Clinic, Tartu University Hospital, Puusepa 8, 50406 Tartu, Estonia

**Keywords:** adolescent health, sprint interval training, metabolic syndrome risk score, adipokines, pediatric obesity

## Abstract

This study investigated the effect of supervised sprint interval training (SIT) on different cardiometabolic risk factors and adipokines in adolescent boys with obesity. Thirty-seven boys were allocated to either a SIT group (13.1 ± 0.3 years; body mass index [BMI]: 30.3 ± 0.9 kg·m^−2^) or a control group (CONT) (13.7 ± 0.4 years; BMI: 32.6 ± 1.6 kg·m^−2^). The SIT group performed 4–6 × 30 s all-out cycling sprints, interspersed with 4 min rest, for 3 sessions/week, during a 12-week period, while the non-exercising CONT group maintained a habitual lifestyle. Anthropometric measurements, triglycerides, fasting insulin and glucose, total cholesterol (TC), high- (HDLc) and low-density (LDLc) cholesterol, leptin and adiponectin in blood, cardiorespiratory fitness (CRF), and a metabolic syndrome severity risk score (MSSS) were calculated before and after the 12-week period. Compared to baseline values, a significant reduction in MSSS was seen in the SIT group after intervention. LDLc showed favorable changes in SIT compared to CONT (−0.06 ± 0.1 vs. 0.19 ± 0.01 mmol·L^−1^; *p* = 0.025). Additionally, CRF increased in the SIT group compared to the CONT group (5.2 ± 1.1 vs. −2.1 ± 1.1 mL·min^−1^·kg^−1^, *p* < 0.001). Moreover, a 12-week all-out SIT training effectively improves cardiometabolic health in adolescent boys with obesity.

## 1. Introduction

Obesity during growth and maturation is a serious global health problem, and the prevalence of childhood and adolescent obesity has increased tenfold during the last four decades [1]. Obese youth are at risk of developing cardiovascular disease, insulin resistance, type 2 diabetes, and other orthopaedic, pulmonary, and physiological disorders later in life [2,3]. If this health concern is not addressed promptly during growth and maturation, obesity in adulthood causes high costs to health care systems and puts an immense socio-economic burden on individuals [4]. For example, in the context of the cost of obesity, a systematic literature review revealed that the average 3-month health care costs per person in Germany are estimated to be 1244 euros [4]. This knowledge demonstrates an urgent need to develop effective strategies to reduce obesity in young people and improve overall health across their lifespan.

It appears that time-efficient high-intensity interval training (HIIT) can increase cardiorespiratory fitness (CRF) and improve body composition among young and adult populations [5,6,7,8]. However, sprint interval training (SIT), a more intense and shorter form of HIIT, is still underused among adolescents with obesity [9]. SIT has shown a favorable effect on CRF and body composition in adolescent boys and girls [10,11], but as important as the enhancement of CRF and body fat loss, changes in cardiometabolic biomarkers are also important factors in the improvement of health in growing and maturing youth. Several well-established blood biomarkers, such as glucose, insulin, total cholesterol (TC), triglyceride (TG), low- (LDLc) and high-density (HDLc) lipoprotein cholesterol, and insulin resistance (IR) have been used to evaluate possible risks to human health and are recognised as cardiovascular disease (CVD) risk factors in adolescents [12]. For instance, a rise in TG and LDLc or a decrease in HDLc levels increases the risk of atherosclerosis and the prevalence of coronary heart disease [13,14] and predicts future type 2 diabetes [15]. Although physiological IR usually occurs during puberty, obesity during adolescence aggravates it [16]. Another significant CVD and type 2 diabetes risk indicator for later life is the presence of metabolic syndrome (MetS) in childhood [17]. Low CRF is associated with higher MetS [18,19], and elevated MetS corresponds with higher levels of LDLc in adolescents [20]. An increase in CRF through adolescence to young adulthood is positively related to HDLc and negatively associated with a change in blood pressure, IR, TG, and insulin [19]. Lately, a sex- and racial/ethnicity-specific MetS severity z-score (MSSS) has been developed as a continuous score to assess the severity of metabolic derangement [21], and the successful use of MSSS among youth has been seen in previous studies [17,20,22]. In addition, adipokines such as leptin and adiponectin exert effects on energy homeostasis, regulate lipid and insulin/glucose metabolism, and influence the immune system [23,24,25]. These adipokines are interdependent during weight gain, where the level of the pro-inflammatory marker, leptin, increases and an anti-inflammatory marker, adiponectin level, diminishes [25,26]. Furthermore, adiponectin levels are lower in adolescents with obesity than in non-obese peers [27] and higher in individuals who are regularly involved in intense weight-bearing exercises [28]. Therefore, obesity may lead to another pathological condition, such as chronic systemic low-grade inflammation [24,29,30], which, in turn, may lead to the development of systemic IR [25].

In regard to these aforementioned cardiometabolic health indices and exercise training methods, the findings are still contradictory. Several studies have found that SIT produces beneficial cardiometabolic changes [31,32], while others have reported only some effects [8,33,34,35]. Lipids and adiponectin play a role in the process of how physical activity affects a person’s biological systems [29]. However, findings on how leptin and adiponectin interact together and how different SIT protocols impact the health status of adolescent boys with obesity are still not evident and, therefore, require further studies. Reportedly, 30 s all-out SIT protocol (6 × 30 s all-out cycling sprints with 4 min recovery between sets) has not been used in adolescent boys with obesity [9]. The aim of the present study was to investigate the effect of a 12-week all-out SIT protocol on cardiometabolic-related outcomes, such as fasting serum insulin and glucose concentrations, IR, blood lipids, leptin and adiponectin levels in the blood, and CRF. It was hypothesized that SIT is an effective training strategy to evoke cardiometabolic health benefits among adolescent boys with obesity.

## 2. Materials and Methods

### 2.1. Participants and Study Design

Healthy 12- to 16-year-old adolescent boys diagnosed with childhood obesity (E66.0 by ICD-10) at the Tartu University Children’s Clinic since 1 January 2016 and who presented a body mass index (BMI) above the 95th percentile on the Estonian BMI chart [36] were recruited from Tartu City or Tartu County, Estonia. All participants underwent a full medical screening prior to participation. None of the study subjects attended any regular physical training (except obligatory school physical education lessons twice per week), used any therapies and medications for obesity, or presented any chronic disease that may prohibit testing and the study’s progress. After the recruitment process, eligible participants were distributed into sprint interval training (SIT) or untrained control (CONT) groups. The SIT group underwent baseline testing (Pre), was assigned to a 12-week training intervention period, and was tested again post-intervention (Post). The CONT group participated only in the Pre and Post testing without training intervention. One week before and immediately after the 12-week study period, all participants performed identical testing procedures with identical equipment. During the study, all participants were asked to maintain their usual diet. Each participant and the participant’s legal guardian were informed of the study protocol and testing procedures, and both signed their formal written consent. The study protocol followed the ethical standards of the Helsinki Declaration and was approved by the Medical Ethics Committee of the University of Tartu, Estonia (Consent No 282/T-5 issued on 21 May 2018).

### 2.2. Anthropometry and Pubertal Maturation

The participants’ body mass (A&D Instruments Ltd., Abingdon, UK) and height (Martin metal anthropometer; GPM Anthropological Instruments, Zurich; Switzerland) were measured to the nearest 0.05 kg and 0.1 cm, respectively. Body mass index (BMI, kg·m^−2^) was calculated as body mass in kg divided by body height in meters squared. Waist circumference (WC, in cm), as a parameter of central adiposity, was measured according to the protocol recommended by the International Society for the Advancement of Kinanthropometry [37]. Total body fat mass (FM; in kg and %) and total lean body mass (LBM; in kg) were measured by dual-energy X-ray absorptiometry (DXA; Hologic Discovery QDR Series, Waltham, MA, USA). Coefficients of variations (CVs) for body composition measurements were less than 2%. The pubertal stage was evaluated according to the Tanner method, which uses self-assessment of genitalia and pubic hair stages and has been used in our previous studies with obese boys [38,39].

### 2.3. Cardiorespiratory Fitness

All participants completed the incremental exercise test to exhaustion. An electrically braked bicycle ergometer (Lode Cordival V3, Groningen, Netherlands) was used for exercise testing. The initial workload was set at 50 W, and every 2 min, the workload was increased by 25 W until exhaustion [40]. During the maximal exercise test, respiratory gas exchange was measured breath-by-breath, with data being recorded in 10 s intervals to measure oxygen consumption (VO_2_) by a gas analyzer (SpiroPro SensorMedics, Yorba Linda, CA, USA). Peak oxygen consumption (VO_2_peak; L·min^−1^) was considered the highest VO_2_ rate achieved within 30 s at the end of the exercise test [40]. CRF was defined by absolute VO_2_peak (L·min^−1^), relative VO_2_peak per kg of body mass (VO_2_peak/_kg_; mL·kg^−1^·min**^−1^**) and by VO_2_peak per kg of LBM (VO_2_peak/_LBM_; mL·kg^−1^·min**^−1^**) [38,41]. Heart rate (HR) was continuously monitored with a heart rate monitor (Polar Vantage, Polar Electro OY, Kempele, Finland).

### 2.4. Cardiometabolic Risk Factors

Resting systolic (SBP) and diastolic (DBP) blood pressure was measured in the sitting position from the non-dominant arm as a mean of 3 consecutive measurements at 5 min intervals using a validated auscultatory method (Hillrom Welch Allyn Durashock Aneroid Blood Pressure BP Monitor–DS44, Welch Allyn, New York, NY, USA). The mean of the two closest BP readings was used in further analysis. Mean arterial pressure (MAP, mmHg) was calculated as [(SBP-DBP)/3] + DBP. Blood samples were taken from an antecubital vein between 8:00 and 9:00 AM after an overnight fast. The serum was immediately separated and frozen at −80° until further analysis. Total cholesterol (TC, mmol·L^−1^), triglycerides (TG, mmol·L^−1^), and high- (HDLc, mmol·L^−1^) and low-density (LDLc, mmol·L^−1^) lipoprotein cholesterol levels were measured by conventional methods. Insulin (μU·mL^−1^) was analysed using Immulite^®^ 2000 (Diagnostic Products Corporation, Los Angeles, CA, USA), while concentrations of glucose (mmol·L^−1^) were measured with a commercial kit (Boehringer, Mannheim, Germany). Insulin resistance (IR) was calculated by the Homeostatic Model Assessment (HOMA-IR), which is computed as follows: (fasting serum insulin × fasting serum glucose)/22.5 [42]. To define IR in pubertal boys, a HOMA-IR cut-off value of 5.22 was used (sensitivity 56%, specificity 93.3%) [43]. In addition, leptin (ng·mL^−1^) was determined by radioimmunoassay (Mediagnost GmbH, Reutlingen, Germany) with the least detection limit of 0.01 ng·mL^−1^, and the intra- and inter-assay CVs less than 5%, while adiponectin (μg·mL^−1^) was also determined by commercially available RIA kit (Linco Research, St. Charles, MO, USA) with the least detection limit of 1 µg·mL^−1^, and the intra- and inter-assay CVs less than 7% [39]. A metabolic syndrome severity risk score (MSSS, z-score) was formed by multiplying each participant’s clinically measured values (height, body mass, WC, SBP, HDLc, TG, and fasting glucose) by subgroup-specific coefficients. For calculating MSSS, the Pediatric MSSS online calculator was used (http://mets.health-outcomes-policy.ufl.edu/calculator/, accessed on 2 March 2022) [21]. A lower MSSS indicates a healthier overall risk profile.

### 2.5. Training Program

SIT group performed a 12-week supervised sprint interval training program (3 sessions/week on Monday, Wednesday, and Friday) on an electronically-braked cycle ergometer (Wattbike Pro/Trainer, Vermont House, Wilford Ind Est, Nottingham, England). During non-training days, participants continued with their habitual everyday life. All training sessions consisted of 10 min warm-up, followed by 4–6 repetitions of 30 s all-out cycling bouts interspersed with 4 min active rest after each bout. Finally, training ended with a 5 min cool-down period. Each training session was monitored using a Polar H7 heart rate monitor (Polar Electro OY, Kempele, Finland). The cycling load during the 30 s all-out sprint was self-selected by each participant, but to avoid poor pacing during the sprint phase, the target cycling cadence was at or above a 90 pedalling rate, and the participants were asked to maintain the maximal possible effort during the sprint bout. The number of cycle sprints was increased after every 4 weeks by one additional 30 s cycle sprint to maintain the desired exercise intensity over the 12-week period. Exercise intensity during training was reported via % of maximum HR, mean and peak power output, and fatigue index [(peak power–mean power)/peak power]. The progressive increase of the training load is shown in Table 1. All SIT sessions were supervised, and the participants were verbally encouraged to produce maximal effort to ensure compliance with SIT protocol.

### 2.6. Statistical Analysis

Analyses were performed with the SPSS software (version 21.0; SPSS Inc, Chicago, IL, USA). Standard preliminary procedures were used to test the violations of assumptions. The normality of raw data and residuals from one-way analysis covariance (ANCOVA) was assessed with the Shapiro-Wilk test or through visual examination of histograms and normal quantile-quantile plots. To limit the influence of outliers, raw scores in the data set were first winsorized as used before [44]. The aforementioned value modification method replaces extreme values with the largest or second smallest value in the cohort without discarding outliers [45]. A *t* test for independent samples was used to compare baseline characteristics of the participants between groups at baseline. Fisher’s exact test was used to compare dependent proportions (pubertal maturation). A *t* test for paired samples was used to check possible cardiometabolic and adipokine changes from Pre- to Post-intervention within groups. The intervention’s effects on cardiometabolic profile and adipokine values between the groups were studied by ANCOVA, including group as a fixed factor (SIT, CONT), Post–Pre-intervention change as the dependent variable, and baseline values of the dependent variable as covariates. In addition to baseline values, age and body FM were also used as covariates. To characterize the magnitude of the difference between the two groups, effect size (ES) partial eta squared (*η*^2^) was reported and considered small if *η*^2^ < 0.06 and large if *η*^2^ > 0.14 [46]. As this is a secondary analysis from a parent study, a priori sample size was not possible to calculate, but based on previous studies [6,47] with similar research questions, a sample size of 14 participants per group was estimated. Data are presented as means and standard errors unless otherwise indicated. An *alpha* value of *p* < 0.05 was set as the minimal level of significance.

## 3. Results

### 3.1. Baseline Characteristics

Baseline anthropometric characteristics are reported in Table 2. Groups were similar (*p* > 0.05) in age, body mass, height, and sexual maturation status and exhibited similar adiposity evaluated by BMI, FM (kg and %), and WC but were different (*p* < 0.05) based on some blood markers (Table 2).

### 3.2. Body Composition

The 12-week study induced no between-group differences in any parameters of body composition. Compared to baseline, the CONT group subjects weighed 99.3 ± 6.4 kg before and 102.0 ± 6.5 kg after (an increase of 2.8 ± 0.7 kg, *p* = 0.002), while a non-significant increase by 1.2 ± 0.7 kg was observed in the SIT group. The training program induced a decrease in BF% in the SIT group from 41.1 ± 1.3% to 39.2 ± 1.5% (a decrease of 1.2 ± 0.6%, *p* = 0.006), while a non-significant change from 39.6 ± 1.9% to 38.9 ± 1.9 (an increase of 0.5 ± 0.6%, *p* = 0.170) was marked in the CONT group over time. After the study period, WC was not altered in either group, but the SIT group revealed smaller changes (−0.01 ± 1.7 cm), while an opposite trend was observed in the CONT group (an increase of 3.1 ± 1.7 cm).

### 3.3. Cardiometabolic Risk Factors

Pre, Post, and change (Post-Pre) cardiometabolic profiles and adipokine values are summarized in Table 3. In ANCOVA analysis, all the models were adjusted for baseline levels of the outcome studied, and after a 12-week period, there were no changes between the two groups on the lipid profile or adipokine variables studied, except for LDLc, which was reduced in the SIT compared to the CONT group [F(1,25) = 5.71, *p* = 0.025, *η*^2^ = 0.19]. There were no group differences in leptin and adiponectin levels following the 12-week study period, but a significant difference between groups was observed in CRF parameters, with SIT showing a significantly greater post-intervention increase in absolute and relative VO_2_peak per kg of body mass and per kg of LBM than the CONT group [F(1,25) = 4.52, *p* = 0.044; F(1,25) = 10.02, *p* = 0.004; and F(1,25) = 23.08, *p* < 0.001, respectively]. An additional two covariates, age and body FM change, did not alter the ANCOVA results (data not shown). Compared to baseline values, HDLc and adiponectin showed a significant decrease in both the SIT and CONT groups after the study, but MSSS revealed a significant decrease over time in the SIT group only.

## 4. Discussion

This study evaluated the effect of a 12-week SIT program on cardiometabolic health parameters in obese adolescent boys. The main findings were that a 30 s all-out SIT program evoked favorable changes in the MSSS and improved CRF and BF% after the intervention period.

It is noted that MSSS is related to future type 2 diabetes and CVD risk [17,20]: every one-unit elevation in childhood MSSS increases the odds of type 2 diabetes nearly three-fold, and the risk of CVD nearly 10-fold by 38 years of age [48,49]. Such knowledge may provide a good foundation for the use of sex- and racial/ethnicity-specific MSSS more often in clinical settings or physical therapy programs as it may better reflect the individual’s overall cardiometabolic risk status than following changes in multiple cardiometabolic biomarkers [22]. Therefore, improved (*p* < 0.05) MSSS after 12 weeks of SIT shows training therapy effectiveness among obese adolescent boys. Further, CRF is also associated with MetS [18,19,50]. In the present study, the 30 s all-out SIT increased CRF by 7.1% (*p* = 0.035) compared to the baseline value. These findings are in line with the results from the systematic review [9], suggesting that HIIT enhances CRF in obese adolescents. Therefore, not only the improvement in MSSS but also an enhancement in CRF could be a key to reducing CVD deaths later in life [51].

After the 12-week study, a large and significant difference (*p* = 0.025, *η*^2^ = 0.19) between the two groups was observed only in the LDLc variable. Although the within-group change after the study did not reach statistical significance in either group, the between-group change observed in the current study may suggest a health benefit value as the reduction of pro-inflammatory marker LDLc even by 1% decreases the prevalence of coronary heart disease by 2% [52]. Previously, different studies reported positive alterations in lipid profile in response to intermittent training together with weight loss and changes in body composition in adolescents with obesity [32,53,54], but the others have not shown the effects of training on lipids [11,55,56]. Accordingly, unchanged LDLc levels were observed even after 12 weeks of 2 min of running at 80–90% VO_2_max interspersed with 1 min recovery among obese adolescent boys (mean age of 13.0 ± 0.8 years) with simultaneous positive changes in body composition and TG levels [53]. In contrast, a study conducted by Ouerghi et al. [8] showed that after 8 weeks of 16 × 30 s runs at 100–110% maximum aerobic speed with a recovery of 30 s induced a significant decrease in LDLc concentration, body composition parameters, and TG and TC levels among overweight/obese young men (17–20 years). The levels of blood lipids are dependent on age, gender, physical activity, and body fatness [57]. Further, studies with obese adolescents have shown an increase [32,53] or no change [8,11,56] in HDLc after different HIIT protocols. In the present study, intriguingly, the post-intervention levels of HDLc decreased in both groups (*p* < 0.05) compared to baseline values. One explanation could be the age factor, as both groups were at the mean age of 13.1 ± 1.3 years for SIT and 13.7± 1.6 years for CONT. It appears that HDLc levels follow certain age- and sex-related patterns where it decreases up to 16 years in boys and up to 14 years in girls [57], especially in obese individuals [27]. For instance, increased HDLc levels after 12-week running-based HIIT were observed among obese girls with a mean age of 15.6 ± 0.7 years [32]. Still, the current study’s finding contradicts Koubaa et al.’s [53] study population (obese adolescent boys with a mean age of 13 ± 0.8 years), whose HDLs significantly increased after 12-week running-based HIIT. These investigations confirm a large controversy regarding the impact of SIT on lipid profile and support the fact that different demographic characteristics (age, gender), training intervention protocols, and changes in body composition may be the confounding factors in how different metabolic biomarkers change on each individual after intermittent training.

It has been reported that HOMA-IR correlates with age as it increases from 10 to 13 years of age in obese adolescent boys and, afterward, starts decreasing again [58]. Although the groups did not significantly differ in age at baseline, it was seen that the CONT group was 0.6 years older, and individual observation showed that there were more boys in Tanner 5 stage compared to the SIT group (6 vs. 1, respectively). Accordingly, it was speculated that the CONT group may have had their physiological rise in HOMA-IR level due to their Tanner stage [43]. Following the 12-week study period, both groups reduced fasting glucose, insulin, and HOMA-IR scores compared to baseline values, but none of the changes was significant. In clinical research, reliable HOMA-IR is a frequently used method in obese children and adolescents for determining insulin resistance [58], and even a small reduction in body fat is effective in improving IR [59]. Therefore, it was surprising that after a significant loss in BF% in the SIT group after the training period, the HOMA-IR index remained unchanged. Similar to the current study’s results, Morrisey et al. [60] found that 12 weeks of 4–6 120 s shuttle runs at 90–95% HR_max_ with active rest favorably altered body composition, while fasting insulin and HOMA-IR scores stayed unchanged in obese adolescent boys and girls. In contrast, Racil et al. [54] found that the positive changes in body composition after 12-week SIT training resulted in a decrease in insulin levels and HOMA-IR scores in obese adolescent females. Despite unchanged fasting insulin, glucose, and HOMA-IR values in the present study, there was a significant reduction in BF% (*p* = 0.006) compared to baseline in the SIT group making this finding similar to the study by Eddolls et al. [61] by showing HIIT as an effective method for improving body composition in adolescents.

A decrease in adiponectin levels after the 12-week study period was observed in both groups. These findings are controversial to Racil et al.’s [32] results, where a reduction of adiponectin was seen after 12-week HIIT training. It has been described that adiponectin declines throughout puberty, with the lowest levels among obese pubertal boys compared to obese girls in puberty and post-puberty [27]. The study population in Racil et al. [32] consisted of girls at a mean age of 15.9 ± 0.3 years, whose adiponectin levels had already started to physiologically increase due to their pubertal status. Therefore, adiponectin levels change during puberty and may influence boys more than girls. In addition, adiponectin levels are correlated with BMI, insulin, HOMA-IR, and HDLc [27]. Improvements in all aforementioned indices after 12-week HIIT were seen in Racil et al. [32] but not in the present study. This could be another explanation for why no favorable changes in adiponectin levels were found in the current study. Moreover, it could be speculated that the work modality used in SIT protocols may be another factor influencing the differences between the studies. For instance, 60 × 8 s all-out cycling sprints evoked no positive changes in adiponectin levels even after a 15-week SIT training in young women (mean age of 20.2 years) [35], whereas a 12-week total of 12–16 × 30 s shuttle runs at 100–110% maximum aerobic speed among obese girls significantly increased adiponectin levels [32]. The exercise intensity in the Racil et al. [32] study was based on the speed associated with VO_2_peak, which was increased over the 12-week period. In the current study, the employed training intensity was founded on the % of maximum HR, which remained the same during the whole training period. It is assumed that training intensity should be adjusted according to the training progression not only by sprint numbers but also by adjusting exercise intensity throughout the intervention period.

Following the 12-week study, no reduction was seen in fasting leptin in both groups. As the change in fat mass was not different between the groups, the current study did not see significant changes in leptin levels within or between the groups, similar to the study in overweight and obese young women at a mean age of 19.4 years, where no changes were also observed among body composition or any other blood variables [33].

This study has some limitations that should be considered. Initially, the number of subjects in the study groups was small, which may have influenced the statistical power to discover the differences between the groups. However, our number of subjects is consistent with similar studies [6,10,31,53,62]. Secondly, the current study was designed without caloric restriction. Diet as the most important factor in weight management [63] could be considered in further studies to see more positive changes in the cardiometabolic determinants among obese adolescents. Although the attendance rate for training sessions was at least 80%, when planning for future studies, an attendance rate of more than 90% could have a bigger impact on the final results, as seen from several previous studies [10,32,54,62]. The strength of the current study is the existence of a non-exercising control group. It gives a good delineation of the independent effect of the intervention and gives some clarity in interpreting the results. Another strength is that the current study was conducted only among boys. Several previous studies have included both boys and girls [10,11,55,62] or only girls [32,54], which limits the clarity of the results. An important advantage of the current study was that the SIT protocol was carried out on cycle ergometers, which may help prevent the prevalence of traumatic injuries that can be caused by excess body weight overloading the joints while engaging in weight-bearing HIIT exercises such as running or jumping, therefore, making cycling SIT a beneficial training mode in improving the cardiometabolic health in obese adolescents. Although studies among healthy, overweight, and/or obese adolescents [9] have shown the feasible and safe use of SIT, more studies conducted among the clinical population should be carried out before reassuring the safety of SIT among these cohorts [64]. Before the reassurance, performing SIT in an obese population should be addressed with prior medical screening and care.

## 5. Conclusions

In conclusion, all-out intermittent cycling bursts improve such cardiometabolic risk factors as MSSS, CRF, and BF% in adolescent boys with obesity without substantial changes in any other lipids or adipokine levels. However, as no previous studies have used 30 s all-out cycling sprint training protocol among adolescent boys with obesity, further studies are needed before any conclusions can be drawn.

## Figures and Tables

**Table 1 ijerph-19-12672-t001:** Mean (± SE) characteristics of training protocol.

Variable	Week 1–4	Week 5–8	Week 9–12
SIT	4 × 30 s all-out, 4 min recovery	5 × 30 s all-out, 4 min recovery	6 × 30 s all-out, 4 min recovery
HR_max_ (%)	81.7. ± 1.5	82.5 ± 1.5	79.8 ± 1.2
Peak Power (W/kg)	5.8 ± 0.4	6.5 ± 0.6 *	7.0 ± 0.6 *^,#^
Mean Power (W/kg)	3.6 ± 0.3	3.8 ± 0.3 *	3.6 ± 0.3
Fatigue index	38.2	40.1	46.9

SE, standard error; SIT, sprint interval training; HR_max_, maximum heart rate. * Significantly different (*p* < 0.05) from the corresponding values of Week 1–4. ^#^ Significantly different (*p* < 0.05) from the corresponding values of Week 5–8.

**Table 2 ijerph-19-12672-t002:** Mean (± SE) baseline anthropometric characteristics in the sprint interval (SIT) and non-exercising (CONT) groups.

Variable	SIT (*n* = 14)	CONT (*n* = 14)	*p* Value
Age (years)	13.1 ± 1.3	13.7 ± 1.6	0.320
Tanner stages I/II/III/IV/V (%)			0.348
I	0	0	
II	21.43	14.28	
III	21.43	14.28	
IV	50.00	35.72	
V	7.14	35.72	
Body mass (kg)	89.1 ± 15.9	99.3 ± 23.9	0.198
Height (cm)	170.6 ± 10.0	173.5 ± 10.8	0.301
BMI (kg·m^−2^)	30.3 ± 3.2	32.6 ± 5.9	0.408
FM (kg)	35.9 ± 2.1	38.6 ± 3.4	0.506
BF (%)	41.1 ± 1.3	39.6 ± 1.9	0.495
WC (cm)	96.2 ± 2.4	100.8 ± 4.3	0.364
MAP (mmHg)	91.7 ± 3.1	93.4 ± 3.7	0.725
VO_2_peak (mL·kg^−1^·min^−1^)	29.7 ± 1.4	27.6 ± 1.6	0.336
Glucose (mmol·L^−1^)	5.19 ± 0.1	5.24 ± 0.1	0.647
Insulin (μU·mL^−1^)	20.91 ± 1.7	34.41 ± 5.0	0.020
HOMA-IR	4.85 ± 0.4	8.11 ± 1.2	0.024
TC (mmol·L^−1^)	3.90 ± 0.1	4.67 ± 0.2	0.005
HDLc (mmol·L^−1^)	1.16 ± 0.06	1.03 ± 0.04	0.084
LDLc (mmol·L^−1^)	1.99 ± 0.1	2.53 ± 0.2	0.014
TG (mmol·L^−1^)	1.12± 0.1	1.89 ± 0.2	0.005
Adiponectin (μg·mL^−1^)	4.11 ± 0.3	3.59 ± 0.3	0.227
Leptin (ng·mL^−1^)	26.49 ± 2.2	28.61 ± 1.6	0.447
MSSS (z-score)	2.16 ± 0.1	2.27 ± 0.1	0.353

SE, standard error; BMI, body mass index; FM, fat mass; BF, percentage of body fat; WC, waist circumference; MAP, mean arterial pressure; VO_2_peak, peak oxygen consumption; HOMA-IR, homeostasis model assessment of insulin resistance; TC, total cholesterol; HDLc, high-density lipoprotein cholesterol; LDLc, low-density lipoprotein cholesterol; TG, triglycerides; and MSSS, metabolic syndrome severity risk score. Differences in pubertal maturation between groups were analyzed using Fisher’s exact test.

**Table 3 ijerph-19-12672-t003:** Mean (±SE) cardiorespiratory and blood chemistry data before (Pre) and after (Post) 12 weeks in sprint interval training (SIT) and control (CONT) groups.

Variable	SIT (*n* = 14)		CONT (*n* = 14)		Difference ^a^ (95% CI) ^b^			
	Pre	Post	Pre	Post	SIT	CONT	*p* Value	*η* ^2^
MAP (mmHg)	91.7 ± 3.1	89.5 ± 2.9	93.4 ± 3.7	92.5 ± 2.9	−2.6 ± 2.2 (−7.2, 2.0)	−0.5 ± 2.2 (−5.1 ± 4.2)	0.509	0.02
VO_2_peak (L·min^−1^)	2.6 ± 0.2	2.9 ± 0.2 ^#^	2.7 ± 0.2	2.8 ± 0.2	0.3 ± 0.1 (0.1, 0.5)	0.1 ± 0.1 (−0.1, 0.2)	0.044	0.15
VO_2_peak/_kg_ (mL·kg^−1^·min^−1^)	29.7 ± 1.4	33.0 ± 1.2 ^#^	27.6 ± 1.6	28.0 ± 1.3	3.7 ± 0.8 (2.0, 5.3)	0.1 ± 0.8 (−1.6, 1.7)	0.004	0.29
VO_2_peak/_LBM_ (mL·kg^−1^·min^−1^)	53.9 ± 2.1	57.5 ± 1.3 ^#^	48.8 ± 1.8	48.1 ± 1.1	5.2 ± 1.1 (3.0, 7.3)	−2.1 ± 1.1 (−4.4, −0.04)	<0.001	0.48
Glucose (mmol·L^−1^)	5.19 ± 0.1	5.11 ± 0.1	5.24 ± 0.1	5.19 ± 0.1	−0.09 ± 0.1 (−0.21, 0.03)	−0.04 ± 0.1 (−0.16, 0.03)	0.558	0.01
Insulin (μU·mL^−1^)	20.91 ± 1.7	20.64 ± 1.8	34.41 ± 5.0	31.80 ± 6.6	−1.63 ± 2.0 (−5.76, 2.51)	−4.33 ± 2.0 (−8.46, −0.19)	0.376	0.03
HOMA-IR	4.85 ± 0.4	4.72 ± 0.4	8.11 ± 1.2	7.45 ± 1.6	0.06 ± 0.6 (−1.20, 1.33)	−0.86 ± 0.6 (−2.12, 0.40)	0.305	0.04
TC (mmol·L^−1^)	3.90 ± 0.1	3.99 ± 0.1	4.67 ± 0.2	4.64 ± 0.2	−0.04 ± 0.1 (−0.22, 0.14)	0.11 ± 0.1 (−0.07, 0.29)	0.272	0.05
HDLc (mmol·L^−1^)	1.16 ± 0.06	1.07 ± 0.05 **^#^**	1.03 ± 0.04	0.96 ± 0.03 **^#^**	−0.06 ± 0.03 (−0.12, −0.003)	−0.09 ± 0.03 (−0.15, −0.04)	0.450	0.02
LDLc (mmol·L^−1^)	1.99 ± 0.1	2.05 ± 0.1	2.53 ± 0.2	2.60 ± 0.1	−0.06 ± 0.1 (−0.21, 0.08)	0.19 ± 0.1 (0.05, 0.34)	0.025	0.19
TG (mmol·L^−1^)	1.12± 0.1	1.20 ± 0.1	1.89 ± 0.2	1.69 ± 0.2	0.03 ± 0.1 (−0.22, 0.27)	−0.14 ± 0.1 (−0.38, 0.10)	0.364	0.03
Adiponectin (μg·mL^−1^)	4.11 ± 0.3	3.49 ± 0.3 **^#^**	3.59 ± 0.3	3.15 ± 0.2 **^#^**	−0.55 ± 0.1 (−0.83, −0.27)	−0.52 ± 0.1 (−0.79, −0.24)	0.865	0.001
Leptin (ng·mL^−1^)	26.49 ± 2.2	26.69 ± 1.2	28.61 ± 1.6	25.12 ± 2.2	−0.23 ± 1.4 (−3.05, 2.59)	−3.06 ± 1.4 (−5.88, −0.24)	0.159	0.08
MSSS (z-score)	2.16 ± 0.1	2.09 ± 0.1 **^#^**	2.27 ± 0.1	2.27 ± 0.1	−0.06 ± 0.03 (−0.11, 0.003)	−0.01 ± 0.03 (−0.06, 0.04)	0.222	0.06

SE, standard error; CI, confidence interval; *η*^2^, eta squared for effect size; MAP, mean arterial pressure; VO_2_peak, peak oxygen consumption; HOMA-IR, homeostasis model assessment of insulin resistance; TC, total cholesterol; HDLc, high-density lipoprotein cholesterol; LDLc, low-density lipoprotein cholesterol; TG, triglycerides; MSSS, metabolic syndrome severity risk score. ^a^ Marginal means from ANCOVA; ^b^ Descriptive values for the differences adjusted by the corresponding baseline values of the outcome; ^#^ Indicate significant change (*p* < 0.05) from baseline.

## Data Availability

The data presented in this study are available on request from the corresponding author for researchers who meet the criteria for access to confidential data.

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
