# Peer review of "Effect of Sprint Interval Training on Cardiometabolic Biomarkers and Adipokine Levels in Adolescent Boys with Obesity"

_ijerph, 2022, doi:10.3390/ijerph191912672_

Round 1
Reviewer 1 Report
The article comprises a relevant issue, but there are many important questions to be solved before it could be considered for publication. The data analysis must be improved, as well as the discussion. Conclusion is not fully supported by showed data.
If the body weight of CONT pre is 99.3kg and post is 102.0kg (lines 204 and 205), resulting in and p-value was 0.002 (with 3kg of difference), how the body mass between CONT (99kg) and SIT (89kg) resulted in a p-value of 0.198 (with 10 kg of difference)? Also, at line 206 to 208: ...”from 41.1 ± 1.3% to 39.2 ± 1.5% (a decrease of 1.2 ± 0.6%, p 207 = 0.006)”. From 41 to 39% p was 0.006. Please, clarify how the comparison for the same parameter between groups showed at table 1 shows p = 0.495. Also, please present the pre and post %BF of CONT at text, with p-value.
It is necessary to show the comparison between groups at baseline for all data. See insulin as an example, with a possible group effect (CONT > SIT). I suggest to use an anova two-way (factorial) and show the F and P of the main effects and interaction. HOMA show the same behavior than insulin, denoting a group difference, something quite relevant for the choose study design and intervention. I also do not fully agree with the results analysis and mainly its interpretation. At first sentence of discussion, we see: …”Wingate-based all-out SIT program evoked favourable changes in the MSSS and in LDLc levels and improved CRF and BF% after the intervention period.”. The table 2 do not show SIT effect on LDLc, for example. Despite lines 216 and 217 describe an effect, both groups showed higher (NS) values after than pre intervention. The minor values are found for SIT, but since the beginning, denoting a GROUP, not TRAINING effect. SIT group seems to be at a lower maturation status, lower body mass, BMI and FM.
Line 216: It would be “except”?
Please, show the VO2 peak in absolute and relative values for whole body and relativized by body lean mass. Such data are important to elicit a full data analysis and its interpretation.
The discussion about your data of adiponectin is not clear and elucidative. Training (both) reduced the adiponectin, as well as HDL.
The conclusion is not support by LDL data showed at table 2.
Reviewer 2 Report
With regard to the manuscript: Effect of sprint interval training on cardiometabolic biomarkers 2 and adipokine levels in adolescent boys with obesity, submitted to Int. J. Environ. Res. Pub-lic Health.
· This is a very interesting manuscript addressing SIT in adolescent with obesity, which is important in context of public health. The paper will contribute to knowledge and is worthy of publication. The manuscript needs minor revision, but the general scope of the study appears to be acceptable. While your work is of interest, allow me to give you a few suggestions
Introduction
· The introduction provides an overview on the background and research question, but the novelty of the study must be highlighted.
· The authors do not explain the reason why they decided to indagate on SIT in adolescent with obesity. Why not was studied the HIIT? Why emphasis is placed on SIT (but not HIIT)?
Methods
· Describe the training program in more detail (Didactics would improve with the inclusion a study design).
· I suggest that the authors include some figure (average data could be showed in results section) with regard to respiratory gas exchange during the maximal exercise test.
· Training was performed on Monday, Wednesday and Friday. what happened the other days?
· From a statistical viewpoint, the authors are correct.
Results
· Please add the values of peak oxygen consumption in liters.min-1
· p-value (and t value) and effect size (ES) must be described in table 2
Discussion
· Line 237 “The main findings were that Wingate-based all-out SIT program evoked favourable changes in the MSSS and in LDLc levels and improved CRF and BF% after the intervention period”. This sentence may be followed by a reminder of performing SIT in obese with care (after all, it's a maximum effort).
· Didactics would improve with the inclusion a figure in discussion section (drawn by the authors) explaining their findings.
· It is well written showing interesting data interpretation. It is important to reader to know why SIT is useful for adolescent with obesity. Thus, the authors should warn readers to be cautious when using SIT (due to the risks). The use of SIT in cycle ergometer could be more highlighted since the SIT in running appears to be more shocking (due to lesions induced by body mass)
Round 2
Reviewer 1 Report
I would like to thank the authors for the according and polite answers. I have just a few points to be solved before the article be suitable for publication.
1) After adding lines 172-179 I realized that the load was not systematically prescribed. This is a quite important! So: a) please inform the employed load (exercise intensity), at least the mean pre- and post-training. It will also elicit to show stats for such parameter. I expect a large improve in load and exercise intensity between pre and post, once the sample is not trained and VO2 max was increased; b) Was used the same load for the 4 bouts at each day? Or the participants were permitted to reduce it if wanted? Please confirm it at text. It is important to give the reader more information regarding the executed training; c) Is it possible to show the peak and mean power and fatigue index of the first bout at pre- and post-training? d) I am not sure if the use of the term “Wingate-based SIT” is the most adequate once the load employed was not the same. Please observe it and modify at text if necessary.
2) At lines 310-311: “In the current study, Tanner stage 5 was more observed in CONT compared to SIT (35.7% vs 7.1%, respectively).” However, at line 206 the reader can see an affirmative based on stats that “groups were similar in age and sexual maturation status”: a) Tanner stages at table 1 were not statistically tested (there is no p-value at table), so, the “sexual maturation status” wrote at line 206 cannot be associated to P<0.05). It is necessary to adjust the text; b) The discussion uses age (line 305) as an argument to explain HOMA result. In sequence, also use pubertal status to reinforce such argument. Such situation can be seemed for other parameters along with discussion. However, your stats outcomes do not confirm it (see lines 205-207 and table 1). So, sometimes it is sounding contradictory and the argument not strong enough; c) similar argument is used on age about HDLc discussion. My main concern is that the exercise intensity employed was not observed by the authors. After show the employed load along training, such condition could be observed and used to explain some results. Was the intensity really like Wingate? It is more important to be informed due to the lack of blood lactate concentration or VO2 data during or immediately after some training section. The reader cannot be certain about how the intensity of the exercise was. Discussion about adiponectin, that text was not improved after modifications at lines 330-360, could benefit when analyzing the exercise intensity of the executed training.
